# Semantic supervision purely through 3D shape descriptors

**Helia Ghasemi**[1] (iD)                    HELIA.GHASEMI@STUDENT.UVA.NL
[1] *Universiteit van Amsterdam, The Netherlands*
**Ioana Simion**[1,2,3] (iD)                    I.SIMION@UVA.NL
**Clara I. Sánchez**[1,2,3] (iD)                    C.I.SANCHEZGUTIERREZ@UVA.NL
**Hoel Kervadec**[1,2,3] (iD)                    H.T.G.KERVADEC@UVA.NL
[2] *qurAI group, Informatics Institute, Universiteit van Amsterdam, The Netherlands*
[3] *Amsterdam University Medical Center at the University of Amsterdam, department of Biomedical Engineering and Physics, Amsterdam, The Netherlands*

**Keywords:** 3D supervision, shape descriptors, anatomical priors

## 1. Introduction

In semantic segmentation, annotations for supervision have always been costly to obtain, even through semi-automated tools. Recent advances in promptable models have significantly accelerated this process (Kirillov et al., 2023; Ravi et al., 2024; Isensee et al., 2025), but they still exhibit limitations and remain sensitive to user interactions (Magg et al., 2026), while remaining infeasible to exhaustively verify manually at high resolutions. At the same time, the fundamental limitation of voxel-wise annotation remains: annotations done for a single scan are at best difficult, if not impossible, to reuse on another scan, even from the same patient.

In (Kervadec et al., 2021), the authors question the standard formulation of semantic segmentation as a pixel-wise classification task, in which each pixel is supervised independently. Instead, they propose representing supervision through a small set of high-level shape descriptors, encouraging the model to capture global structure rather than memorize local pixel patterns. This perspective better reflects how humans process images: first forming a high-level understanding, before refining local details. Empirically, they show that this formulation can drastically reduce supervision—from roughly 65,000 labeled pixels per slice to only 16 descriptors in a 5 class setting—while maintaining comparable performance. Extending that work in 3D is straightforward from a formulation perspective, but requires some considerations from an engineering point of view: as shape supervision requires the whole region to be processed at once in order to compute meaningful descriptors, in 3D this means processing the whole scan as a single patch. Despite eventual memory limitations, 3D shape descriptors have the potential to be derived from other forms of prior knowledge (such as textbooks, radiological reports) and reusable across scans, in stark contrast to voxel-wise annotations.

This work presents i) an adapted memory-efficient architecture that can process a 3D scan as a single patch, even at high resolution, ii) a demonstration that this architecture can successfully be supervised *purely* from 3D shape descriptors, which can be used as a platform to conduct future research on shape supervision.

## 2. Methods

**Network architecture**  Adapted from Wang et al. (2022), which is originally a two-stage architecture (from coarse to fine), we keep only the first stage. The size of the patches used is significantly increased to fit the whole scan, and the supervision adapted to the shape supervision paradigm (Figure 1).

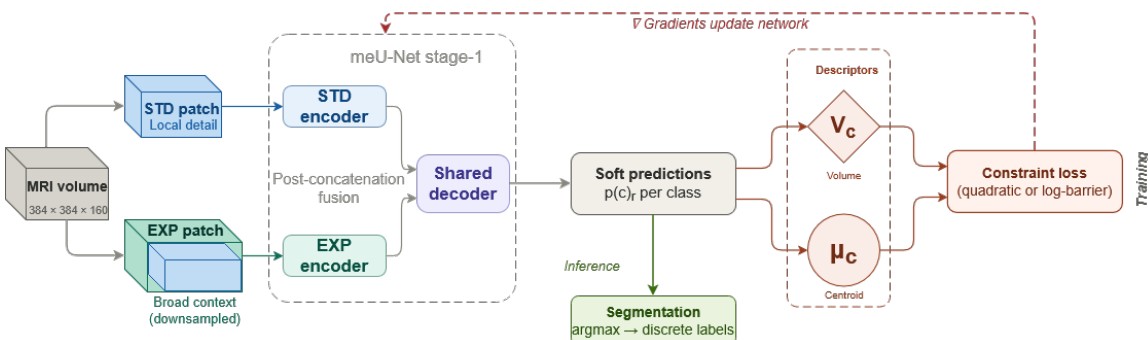

Figure 1: Training and inference pipeline. A volume is sampled into a standard patch (STD) for local detail and an expanded patch (EXP) for broader context. Both streams are fused to predict per-class probabilities.

**3D shape descriptors**  For a segmentation $s$, general 3D shape moments are parametrized by $p$, $q$, $r$ and form the backbone of the shape descriptors used in this paper:

$$m_{p,q,r}(s;k) = \sum_{i \in \Omega} s(k,i) x_i^p y_i^q z_i^r \quad \in \mathbb{R}, \tag{1}$$

with $i = (x_i, y_i, z_i) \in \Omega$ the coordinates from image-space $\Omega \subset \mathbb{R}^3$, and $s(k,i) \in [0,1]$ denoting the segmentation value of class $k$ at voxel $i$.

From this we can compute directly the per-class *volume* of a segmentation, which is simply $\mathfrak{V}(s;k) = m_{0,0,0}(s;k)$ (the sum of all voxels), and the centroid (average of voxel coordinates): $\mathfrak{C}(s;k) = \left( \frac{m_{1,0,0}(s;k)}{m_{0,0,0}(s;k)}, \frac{m_{0,1,0}(s;k)}{m_{0,0,0}(s;k)}, \frac{m_{0,0,1}(s;k)}{m_{0,0,0}(s;k)} \right)$. As in (Kervadec et al., 2021), we can also compute the *average distance to the centroid* $\mathfrak{D}(s;k) \in \mathbb{R}^3$. Thus, for four classes, $\mathfrak{V} + \mathfrak{C}$ gives $4 \times (1 + 3) = 16$ scalar descriptor targets, while adding axis-wise $\mathfrak{D}$ gives 28.

**Loss and supervision**  Instead of supervising individual voxels as with a cross-entropy loss or Dice loss (e.g., $\mathcal{L}_{\text{CE}}(s_\theta, y) \propto \sum_i \sum_k -y(k,i) \log(s_\theta(k,i))$ with $y$ the label and $s_\theta$ the network predictions), we supervise *only the value of the shape descriptors*, and not individual voxels, through loose bounds of the target value:

$$\mathcal{L}_{\text{shape}}(s_\theta, y) \propto \sum_{\mathfrak{f} \in \{\mathfrak{V},\mathfrak{C},\mathfrak{D}\}} \sum_k \left[ \widetilde{\psi}_t \big( 0.9\mathfrak{f}(y;k) - \mathfrak{f}(s_\theta;k) \big) \right. \tag{2}$$

$$\left. + \widetilde{\psi}_t \big( \mathfrak{f}(s_\theta;k) - 1.1\mathfrak{f}(y;k) \big) \right],$$

with $\widetilde{\psi}_t(z) = \begin{cases} -\frac{1}{t} \log(-z) & \text{if } z \leq -\frac{1}{t^2} \\ tz - \frac{1}{t} \log(\frac{1}{t^2}) + \frac{1}{t} & \text{otherwise,} \end{cases}$, $t = 5$, as the extended log-barrier (Kervadec et al., 2022).

## 3. Experiments

**Dataset & implementation details**  We use sagittal 3D DESS knee MRI from the Osteoarthritis Initiative (OAI) at $384 \times 384 \times 160$ resolution (Peterfy et al., 2008) with (Ambellan et al., 2019) as reference segmentations for four classes: femur, tibia, femoral cartilage and tibial cartilage. We use 406 scans for training and validation, and 101 for testing. A non-maximum suppression post-processing (Isensee et al., 2021) is used. All models are trained on one NVIDIA A100 (40 GB VRAM) or Titan RTX (24 GB VRAM).

**Results**  Despite the substantial reduction in supervision, the shape descriptor-only models recover the overall shape and location of all four anatomical structures (Fig. 2). The bones show the best performance while the cartilages (smaller, thinner structures) have a significantly lower Dice (Table 1). The improvement between Fig. 2c and 2d, from adding the extra descriptor type $\mathfrak{D}$, yielded a +12.4% gain on the Dice, demonstrating that additional shape and anatomical priors can be effectively added to boost performance.

The low performance of the tibial cartilage—predicted as a single, continuous connected component, instead of two—could directly benefit from topology-aware losses (Clough et al., 2020) that would not require extra annotations.

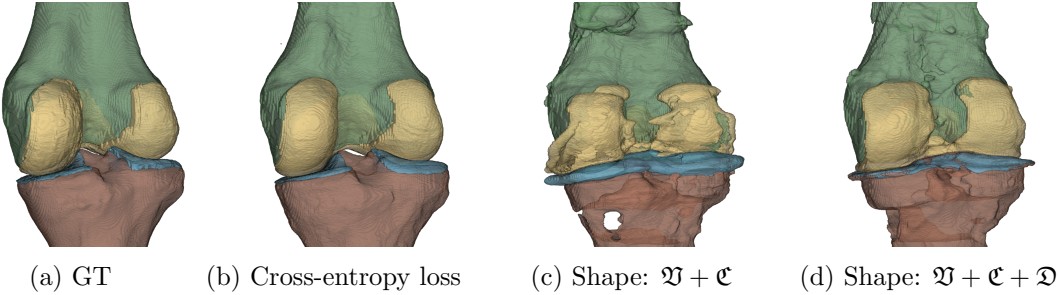

| (a) GT | (b) Cross-entropy loss | (c) Shape: $\mathfrak{V} + \mathfrak{C}$ | (d) Shape: $\mathfrak{V} + \mathfrak{C} + \mathfrak{D}$ |

Figure 2: Qualitative comparison of 3D knee segmentations on the test set.

Table 1: Dice Score (DSC) for different settings across the test set.

| Setting | DSC (%, avg.±std.) | | | | | Supervision |
|---|---|---|---|---|---|---|
| | Avg | Femur | Fem. Cart. | Tibia | Tib. Cart. | |
| Full pixel-wise supervision, Fig. 2b | 92.5 | 98.3±0.5 | 88.0±2.6 | 98.5±0.4 | 84.6±4.5 | 24M voxel labels |
| Volume $\mathfrak{V}$, centroid $\mathfrak{C}$, Fig. 2c | 54.7 | 75.4±29.1 | 43.1±2.6 | 69.5±5.1 | 30.8±3.3 | 16 scalar targets |
| Volume $\mathfrak{V}$, centroid $\mathfrak{C}$, dist. centroid $\mathfrak{D}$, Fig. 2d | 67.1 | 87.0±12.9 | 50.5±2.8 | 78.9±4.0 | 51.9±4.0 | 28 scalar targets |

## 4. Conclusion

We have shown that an adapted 3D-CNN can process a whole high-resolution 3D scan, enabling purely 3D shape-based supervision: reducing supervision from 24 million annotated voxels per scan to as few as 16 scalar descriptor targets, while still producing promising results. Notably, we have shown that additional descriptors reduce the performance gap.

Future work will study more expressive descriptors and robustness to estimated or noisy descriptor targets, rather than descriptors computed from reference masks. Such descriptors could eventually be derived from anatomical priors, reports or partial annotations, and used to train or fine-tune foundation models at very little annotation cost.

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
