# OpenReview forum: "Semantic supervision purely through 3D shape descriptors"
_MIDL.io/2026/Short_Papers — MIDL 2026 - Short Papers Poster_

### Official Review · Reviewer_TjPc · 2026-04-22
**3D shape descriptors is a strong supervision for segmentation**

**Rating:** 4
**Confidence:** 4

**Review:**

The idea is quite interesting as the proposed 3D shape descriptors, i.e., volume, centroid, average distance to the centroid, are just 7 sparse numbers that can be estimated with less annotation efforts compared with dense mask annotation. The segmentation performance of using  3D shape descriptors as supervision is good.

**Summary:**

The paper propose to use 3D shape descriptors, i.e., volume, centroid, average distance to the centroid, as supervision for volumetric medical image segmentation. The advantage is that using 3D shape descriptors can greatly reduce annotation cost. The results show that 3D shape descriptors can achieve a  strong supervision for segmentation.

**Strengths:**

1. The proposed 3D shape descriptors, i.e., volume, centroid, average distance to the centroid, are just 7 sparse numbers that can be estimated with less annotation efforts compared with dense mask annotation.
2. The segmentation performance of using 3D shape descriptors as supervision is good.

**Weaknesses:**

1. The authors used ground truth masks to compute the ground truth 3D shape descriptors. From the results, we cannot tell the performance drop when trained with estimated and inaccurate 3D shape descriptors. The average distance to the centroid may be difficult to estimate although it shows great segmentation performance improvement.
2. Typo: "going from 24 million annotated voxels per scan to 16 shape descriptors''. I think there are just 7 shape descriptors for each class.

**Justification Of Rating:**

The proposed 3D shape descriptors are just 7 sparse numbers that can be estimated with less annotation efforts compared with dense mask annotation. The segmentation performance of using 3D shape descriptors as supervision is good. However, it may be challenging to accurately estimate those shape descriptors in real practice.

---

### Decision · Program_Chairs · 2026-05-08

Accept (Poster)